

# How does perceived heat stress differ between urban forms and human vulnerability profiles? – case study Berlin

Nimra Iqbal[1], Marvin Ravan[1], Zina Mitraka[2], Joern Birkmann[1], Sue Grimmond[3], Denise Hertwig[3], Nektarios Chrysoulakis[2], Giorgos Somarakis[2], Angela Wendnagel-Beck[1]

[1]Institute of Spatial and Regional Planning (IREUS), University of Stuttgart, Stuttgart, 70569, Germany
[2]Remote Sensing Lab, Foundation for Research and Technology Hellas, Heraklion, 70013, Greece
[3]Department of Meteorology, University of Reading, RG6 6ET, Reading, UK

*Correspondence to*: Nimra Iqbal (nimra.iqbal@ireus.uni-stuttgart.de)

**Abstract.**

Urban areas in all world regions are experiencing increasing heat stress and heat-related risks. While in-depth knowledge exists in terms of the urban heat island effect and increased heat stress in cities in the context of climate change, less is known about how individual heat perceptions and experiences differ between urban forms or with different vulnerability profiles of exposed people. It is crucial to identify and assess differences within cities relating to urban form and social structure, as both need to be considered when designing adaptation plans for heat-related risks. Here, we explore linkages between urban structure types (USTs), heat stress perception and different socioeconomic group's experiences in Berlin using a household survey, statistical and earth observation data. We characterize the urban region following the ring structure developed in the *urbisphere* project. Although heat stress exposure is higher in the inner-city ring, we find that a higher percentage of vulnerable groups in the outer city (6 km to 18 km from city centre) where more elderly live. We underscore the need for attention in future adaptation plans based on the USTs, socio-economic profile and adaptive capacities e.g. for elderly living in high-rise buildings with low income and for dense blocks with less green and shaded spaces availability. The method and findings can inform future adaptation strategies of other cities to consider different profiles of vulnerability and adaptive capacities within and between USTs.



## 1. Introduction

Globally, all regions are increasingly affected by climate change (IPCC, 2023). Heat stress is a key challenge  impacting more as urban
citizens increase from the current 56.2% of global population to projected 68.4% by 2050 (United Nations, 2022). While human
vulnerability is highest and resilience lowest in rapidly growing urban areas in developing countries (Birkmann et al., 2016), heatwaves
impact cities globally (e.g., Europe 2003 Schär et al., 2004) highlighting a general need for enhanced resilience. Global increases of near-
surface air temperature are projected to be 2°C by 2050 (IPCC, 2021) without immediate reduction in GHG emissions (Gallardo et al.,
2022). Compound events are likely with urbanization and frequent extreme climate events resulting in adverse consequences (Babiker et
al., 2022). Heat stress impacts urban residents by adding health burdens, notably cardiovascular, respiratory and vector-borne disease (e.g.,
dengue fever and malaria), (IPCC, 2022), and decreasing work productivity (IPCC, 2022). Heat-related mortality, a key climate change
risk to human health ( Vicedo-Cabrera et al., 2021; Lüthi et al., 2023), is exacerbated in urban areas as global and regional temperature
extremes are intensified by the urban heat island effect (Gallardo et al., 2022). Heat risk for individuals depends on temperatures,
exposure, vulnerability and adaptive capacities (Adelekan et al., 2022).

Many studies illustrate impacts of urbanization on heat stress ( Stewart and Oke, 2012; Lemonsu et al., 2015; Narocki, 2021; Tollefson,
2021; Tuholske et al., 2021). With greater urbanization both urban heat islands intensity  (Stewart et al., 2021) and energy consumption
(Voogt and Oke, 2003; Stewart et al., 2021) increase. However, urbanization also plays a pivotal role in reducing the impacts through
climate resilient development (Adelekan et al., 2022) through numerous factors (e.g., urban morphology, vegetation, materials,
anthropogenic heat flux) (IPCC 2020), with urban morphology being one of the strongest influences on urban heat island intensity (Oke,
1981; Grimmond, 2007; Oke et al., 2017b; Gallardo et al., 2022).  Urban form or morphology is impacted by building density and size,
with tall dense buildings have greater re-radiation of longwave radiation therefore retaining heat longer overnight, and reduced airflow
within the urban canopy ( Grimmond, 2007; Oke et al., 2017b). Building materials store large amounts of heat during the day, providing a
large source of energy to be released at night (Grimmond and Oke, 1999; Oke et al., 2017b). Whereas, open vegetated areas can cool more
rapidly at night, facilitating circulation and reducing heat stress. Human activities in domestic, commercial, and industrial areas or traffic-
related heat sources act as a source of anthropogenic heat, contributing to local atmospheric warming (Schwingshackl et al., 2024). These
factors and the inter- and intra-urban spatial disparities can exacerbate exposure and influence  vulnerabilities of disadvantaged urban
dwellers (Adelekan et al., 2022).

Urban and spatial planning primarily focuses on physical urban typologies and phenomena when dealing with climatic risks and adaptation
issues (Turek-Hankins et al., 2021; Wendnagel-Beck et al., 2021; Marando et al., 2022), but different levels of human vulnerability and
adaptive capacities of residents are insufficiently addressed (Turek-Hankins et al., 2021). Despite susceptible group's coping and adaptive
capacity is included in some climate risk assessment frameworks ( Willroth et al., 2012; Birkmann et al., 2013; Kunz-Plapp et al., 2015;
Feldmeyer et al., 2017; Jamshed et al., 2017; Feldmeyer et al., 2019; Zuhra et al., 2019; Sun et al., 2021, Iqbal et al., 2022), this knowledge
is often unconnected in practice (e.g. in climate adaptation plans Hannemann et al., 2023). Heat adaptation plans implemented with
marginalized and vulnerable populations as targets are little published (Eldesoky, A. H. Gil, J. and Pont, 2022), but do include Corburn et
al.'s (2020) tree planting campaign targeting low income areas in Medellín, Colombia. 'Heat equity' of interest, for example, in Paris
(France) involves planning a city-wide network of cooling areas (parks and pools) connected by cool walkways (Nature, 2021) and in
Bochum (Germany) homeless people  are targeted  in their 2021 heat adaptation concept note (Amt für Soziales, 2023). As socio-
demographic and economic aspects of exposed people determines human vulnerability, they are also key when trying to understand and



respond to heat related risk in cities. Thus, urban planning responses to climate change need to better understand dynamics and patterns of
exposure, vulnerability and adaptive capacities of people.

## 1.1. Urban form classification – combining urban morphology and heat characteristics

Urban form and function are important for a wide range of applications in many sectors (e.g. Barlow et al., 2017), including infrastructure
and landscape planning. Form influences many aspects of energy exchange (e.g. Zhou et al., 2011, Oke et al., 2017a; Yue et al., 2019) with
many parameters used to characterise the urban form, for example, floor area ratio and building aspect ratio (Yang et al., 2021; Liu et al.,
2023) and directly impacted by it, for example, sky view factor and shadow fraction.

At the neighbourhood (or local-) scale local climate zones (LCZs) can characterise the areas where near surface air temperature
observations are taken when assessing urban heat island intensity in a globally comparable way (Stewart and Oke, 2012). The LCZ
provide a range of values for each LCZ type for several parameters, including building density, sky view factor and impervious fraction
(Stewart and Oke, 2012). Give the ease of obtaining some of the parameters from satellite-data (e.g. Mitraka et al., 2015; Zhu et al., 2018;
Oliveira et al., 2020) and availability of crowd-source observations urban climate studies have been undertaken both in Berlin (e.g. Fenner
et al., 2017) and in many other cities (e.g. Bechtel et al., 2015; Verdonck et al., 2018; Ren et al., 2019; Aslam et al., 2022). Planners are
using LCZs quite widely (Klopfer, 2023), as LCZ maps of cities are becoming globally available (e.g. Demuzere et al., 2022), but may
lack reliable local expert-generated data for climate adaptation planning use (Klopfer, 2023). With LCZ intended to be global applicable,
some parts of a city may be difficult to classify (Bechtel et al., 2015; Zhu et al., 2018) within the original classes.

City planning departments have combined building metrics (e.g. functional use, number of storeys, building age) to identify urban
structure types (USTs) (Senatsverwaltung für Stadtentwicklung und Umwelt, 2014; Senatsverwaltung für Stadtentwicklung und Wohnen,
2021; LUBW Landesanstalt für Umwelt, Messungen und Naturschutz Baden-Württemberg, 2014)  for use when mapping their regions.
Overall, USTs provide an entry point in analysing inter- and intra-urban variations, both for physical and social urban structures
(Wendnagel-Beck et al., 2021). In climate change studies, USTs have been linked to climate hazards such as heat stress and used for
climate adaptation planning in some cities (LUBW Landesanstalt für Umwelt, Messungen und Naturschutz Baden-Württemberg, 2014;
Senatsverwaltung für Stadtentwicklung, Bauen und Wohnen, 2023). Whilst USTs require expert input and detailed data to be developed
(Klopfer, 2023), as LCZs are intended to use the same "standards" to describe parts of cities they may have greater utility for large scale
applications (Bechtel et al., 2015; Zhu et al., 2018). Nevertheless, both are applicable in a particular city and region, and could be used in
city planning and climate adaptation.

## 1.2. Urban structure type (USTs): considering physical and socio-economic factors to assess cities

Already USTs are important basis of adaptation plans for heat stress in some German cities, with more developing them (Senatsverwaltung
für Stadt-ent-wicklung und Umwelt, 2014; LUBW Landesanstalt für Umwelt, Messungen und Naturschutz Baden-Württemberg, 2014;
Downes et al., 2024). For example, Karlsruhe and Berlin consider USTs in their climate adaptation plans and strategies (Senatsverwaltung
für Stadtentwicklung und Umwelt, 2014; LUBW Landesanstalt für Umwelt, Messungen und Naturschutz Baden-Württemberg, 2014). The
methodology has three steps, that (Wendnagel-Beck et al., 2021): (1) characterizes cities through USTs, (2) identifies climate hotspots that
require adaptation, and (3) develops adaptation measures for different USTs. Many applications have characterized USTs using only
physical indicators (e.g. building age, building height, building use, building geometry, and open space characteristics). In the
identification of climate hotspots, sometimes demographic aspects (e.g. elderly, children and total population density) are captured, for





example, as done in Karlsruhe. However, some key socio-economic and behavioural aspects (e.g., income, risk perception and experience and willingness to adapt) are not investigated fully (Wendnagel-Beck et al., 2021).

USTs are used in urban monitoring; for instance, assessment of peri-urbanization transitions (Downes et al., 2024) and amount of residential greenery (Battisti et al., 2019). UST use in climate assessments includes: thermal performance in Berlin with satellite derived land surface temperature, building height, buildings plan area fractions of and impervious area being influential factors (Kloper (2023),

and indoor and outdoor temperature comparisons in Leipzig (Franck et al., 2013). In Munich, the distance of USTs from the city centre is correlated with land surface temperature (Heldens et al., 2013). A guidebook on adapting to climate change in Dresden uses USTs as an indicator for settlement heat sensitivity (Wende, 2014).

Overall, most studies using USTs focus on the physical structures but lack information on socio-economic and vulnerable populations (e.g. elderly, low income, and/or otherwise disadvantaged groups). The various impacts of heat and perceptions of heat stress for those living in USTs (i.e., detached houses, block development, row houses, large housing estates etc) and their socio-economic attributes (e.g. age,

income) have not been sufficiently explored and integrated into adaptation strategies, despite this information being crucial for effective people centred adaptation. Moreover, in Berlin's Urban Development Plan Climate 2.0 (Senatsverwaltung für Stadtentwicklung, Bauen und Wohnen, 2023) cold air drainage and climate function of open spaces are identified. Less information is given on the population's socio-economic characteristics, their settings, heat stress perceptions, behaviour patterns or adaptation responses. To address these limitations, a household survey (Sect. 2.2.) is undertaken in Berlin to explore:

I.     Does perceived heat stress change from the centre of the urban region towards the periphery?
     II.    How does measured thermal comfort correspond to perceived heat stress by residents?
     III.   How does perceived heat stress differ within an UST and along various USTs?
     IV.    Are the human vulnerability characteristics and adaptive capacity significantly different between USTs or are variances within USTs more significant?
V.     How does perceived heat stress differs amongst various socio-economic groups and vulnerability factors?
     VI.    How can this new knowledge be applied in future climate change adaptation strategies of urban regions?

The ERC *urbisphere* project aims to characterise intra-city variability in a consistent manner globally. To do this, a simple ring structure is identified based on building density and other data sources (Fenner et al. 2024). Here, we capture similarities and differences of perceived

heat, socio-economic structure and adaptive capacities across USTs and city rings in Berlin, and work towards a transferable standardized methodology applicable to other case studies.

## 2. Methods

### 2.1. Berlin study area

Berlin citizens experience heat stress from rising regional temperatures intensified by the urban heat island effect with 1°C increase in

mean annual air temperature between 1971 and 2000 (Deutscher Wetterdienst and Senatsverwaltung für Stadtentwicklung, 2010). Tropical nights (nocturnal air temperature above 20°C) have risen in the inner city by an average of 5 nights between 1967 and 2008 (Deutscher Wetterdienst and Senatsverwaltung für Stadtentwicklung, 2010), and very hot days (maximum day time temperature > 30°C) are expected (Deutscher Wetterdienst and Senatsverwaltung für Stadtentwicklung, 2010) to occur on an average of 25 days annually by 2050



(Deutscher Wetterdienst and Senatsverwaltung für Stadtentwicklung, 2010). With Berlin's continental climate exacerbating summertime
heat, city planning and environmental departments are increasingly keen to enhance their adaptability (Senatsverwaltung für
Stadtentwicklung, Bauen und Wohnen, 2023).

The Berlin region has a polycentric city structure, notably with two city centres existing from the east-west separation after World War
two period. Following the *urbisphere-Berlin* campaign analysis of form and function data (Fenner et al., 2024), identify an inner city ring
(radius 6 km) and an outer city ring (radius 18 km). The latter we split at 12 km, to give three urban rings, which hereafter we refer to as A,
B1, B2 from inner to outer Berlin (Fig. 1a).

The Senatsverwaltung für Stadtentwicklung und Wohnen (2021) identify 13 residential USTs (Table 1) for Berlin. We use socio-
demographic and physical data (Table A1) to reduce this to seven classes (Fig. 1a, Table 1) for comparison. In ring A, there is a larger
proportion of dense and close block USTs (42%) than in either rings B1 or B2 (Fig. 1b). The share of block edge development is also
comparatively higher in the ring A. However, (semi-)detached and terraced houses dominate in rings B1 and B2 (54% and 75%,
respectively). In rings A and B1, row development with landscape green strips are also common (13% and 16%, respectively). Whereas,
large estate development with tower high-rise buildings occur in all three rings but decreasing proportion with distance from the centre (A:
11%, B1: 10% and B2: 8%).

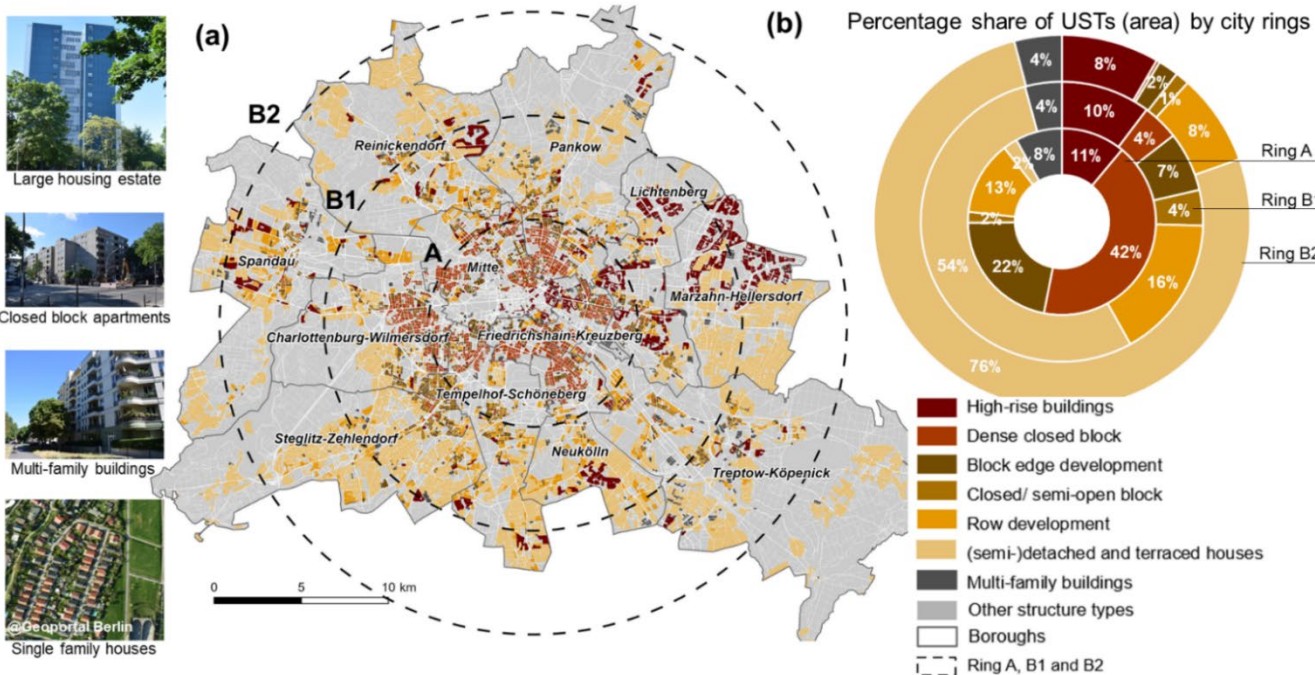

**Figure 1:** Berlin study area (a) inner (A) and outer (B1, B2) city rings and Senatsverwaltung für Stadtentwicklung und Wohnen (2021)
urban structure types (UST, Table 1) with example photos, and (b) plan area of USTs (%) in each city ring. (photo source: Marvin
Ravan)



Table 1: Berlin's urban structure types (UST) (Senatsverwaltung für Stadtentwicklung und Wohnen, 2021), new USTs classes and short name use. Table A1 gives basis for the new classes.

| USTs based on residential form | New Classes | Short name |
|---|---|---|
| Large estate with tower high-rise buildings (1960s-1990s), 4-11-storey | Large estate with tower high-rise buildings (1960s–1990s) | High-rise buildings |
| Dense block development, closed rear courtyard (1870s-1918), 5-6-storey<br>Closed block development, rear courtyard (1870s-1918), 5-6-storey | Dense and closed block (1870s–1918s) | Dense closed block |
| De-cored block-edge development, post-war gap closure (after 1945)<br>Block-edge development with large quadrangles (1920-1940s), 2-5-storey | Block edge development (1920s–post war gap closure) | Block edge development |
| Closed and semi-open block development, decorative and garden courtyard (1870s-1918), 4-storey | Closed and semi-open block development (1870s–1918) | Closed/ semi-open block |
| Free row development with landscaped residential greenery (1950s-1970s), 2-6-storey<br>Parallel row buildings with architectural green strips (1920s-1930s), 2-5-storey | Row development with landscape green strips (1920–1970s) | Row development |
| Densification in single-family home areas, mixed development with yard and semi-private greening (1870s to present)<br>Detached single family houses with gardens<br>Villas and town villas with park-like gardens (mostly 1870s-1945)<br>Row houses and duplex with yards | Detached, semi-detached and terraced houses (1870s–present) | (semi-)detached and terraced houses |
| Rental-flat buildings of the 1990s and later | Different multi-family buildings (1990s–present) | Multi-family buildings |

## 2.2. Household survey and analyses with other data sources

In October 2022, people at 10,000 residential addresses in 39 of the 542 PLRs (Planungsräume or planning areas) (Landesamt für Bürger- und Ordnungsangelegenheiten, 2022) were invited to participate in our household survey (Table 2). PLRs were selected by stratifying across multiple criteria, e.g. heat exposure (Senatsverwaltung für Stadtentwicklung und Umwelt, 2014), population density and

representation of different age groups (Amt für Statistik Berlin-Brandenburg, 2022), unemployment levels (Senatsverwaltung für Stadtentwicklung und Wohnen Berlin, 2019), heat mortality rate (Schuster et al. 2014) to capture a diverse group of people and their behaviour. The 10,000 posted invitations included a QR-code to access the survey online (Evasys GmbH, 2021). With the 565 responses received, all PLR had sufficient responses for analysis, except for one (No 39, 3 responses) being excluded.

The survey has questions on household's heat stress perception and experience, living conditions (e.g. USTs, green space access),
adaptation options, and socio-demographic characteristics (e.g. age, income) (Table 2). The household survey perceived heat stress (percentage of people who said slightly hot to very hot in their neighbourhood, Table 2) is compared to the Senate of Berlin's thermal discomfort index (TDI, Table 3). The TDI uses the Senate of Berlin's Climate Planning Information maps of daytime (14:00) physiological equivalent temperature (PET, (Höppe, 1999)), nocturnal (04:00) air temperature, local characteristics (e.g. plan area of trees (%) and building volume density ($m^3$ $ha^{-1}$)) to assign each block to one of four TDI classes (Table 3) (Senatsverwaltung für
Stadtentwicklung und Wohnen, 2015). We assign each TDI class a value (Table 3) allowing aggregation to PLR scale using area weighted mean.





Table 2: Survey questions analysed in this study (original survey number, Q#) with number of respondents (N), that number as a percentage of PLR respondents or USTs (Respond.). Data availability given in Iqbal et al., 2024.

| referred to as | Question asked of respondents | Respond. (%) | Q# | N |
|---|---|---|---|---|
| Perceived heat | *How hot or cool do you think your neighbourhood is during a heatwave compared to the average outdoor temperature for the city?* <table><tr><td>Much cooler</td><td>Slightly cooler</td><td>No difference</td><td>Slightly hotter</td><td>Very hot</td></tr></table> | % of PLR/ USTs respondents | 5.3 | 558 |
| Housing typologies | *I live in ...* <table><tr><td>Detached single family house</td><td>Semi-detached or terraced single-family house</td></tr><tr><td>Duplex house</td><td>Apartment in a detached multifamily house</td></tr><tr><td>Apartment in an apartment block (covering part of a floor)</td><td>Apartment in an apartment block (covering whole floor)</td></tr><tr><td>Row block building</td><td>Apartment in a multi-family house built in series (block edge development)</td></tr><tr><td>Others</td><td></td></tr></table> | % of USTs respondents | 6.2 | 561 |
| Open spaces | *How would you describe the area right next to your house/apartment?* <table><tr><td>Lots of green (trees, meadow, lawn) and plenty of space between the buildings</td><td>Lots of green (trees, meadow, lawn), but little space between buildings</td></tr><tr><td>Little green (trees, meadow, lawn) and a lot of space between the buildings</td><td>Little green (trees, meadow, lawn), and little space between the buildings</td></tr><tr><td>None of this applies to my living environment</td><td></td></tr></table> | % of PLR/ USTs respondents | 9.1 | 543 |
| Age groups | How old are you? <table><tr><td>18 to 24 years</td><td>25 to 34 years</td><td>35 to 44 years</td><td>45 to 54 years</td></tr><tr><td>55 to 64 years</td><td>65 to 74 years</td><td>75 to 84 years</td><td>85 years and older</td></tr></table> | % of PLR/ USTs respondents | 14.1 | 564 |
| Health Condition | *Have you already had problems with heat stress? If yes, which ones:* <table><tr><td>Lethargy/fatigue</td><td>Trouble sleeping</td><td>Difficulties in concentrating</td><td>Dizziness</td></tr><tr><td>Nausea</td><td>Cardiovascular problems</td><td>Heat stroke</td><td></td></tr></table> | % of PLR respondents | 5.9–5.16 | 559 |
| Household income | *What is the monthly net income (Netto) of the household? (Netto = after deduction of taxes, social security contributions, etc.)* <table><tr><td>Less than 900 €</td><td>900 to under 1300 €</td><td>1300 to under 1700 €</td></tr><tr><td>1700 to under 2000 €</td><td>2000 to under 2300 €</td><td>2300 to under 2600 €</td></tr><tr><td>2600 to under 2900 €</td><td>2900 to under 3200 €</td><td>3200 to under 3600 €</td></tr><tr><td>3600 to under 4000 €</td><td>4000 to under 4500 €</td><td>4500 to under 5000 €</td></tr><tr><td>5000 to under 6000 €</td><td>6000 to under 7000 €</td><td>7000 € and above</td></tr><tr><td>Not specified</td><td></td><td></td></tr></table> | % of PLR/ USTs respondents | 17.8 | 555 |
| Adaptive measures | *Which of the following measure to protect against heatwaves have you already implemented or are you planning to implement (considering the change of weather in Berlin, as described)?* Air conditioner installation <table><tr><td>Already implemented</td><td>In plan/ implementation</td><td>Will be an option for future</td></tr><tr><td>Neither today, nor future</td><td>Does not apply</td><td></td></tr></table> | % of PLR respondents | 12.4 | 369 |

The weighted mean thermal discomfort of PLRs (Table 3) is compared with perceived heat (Table 2) from the survey using Pearson correlation. To understand what may influence perceived heat stress, Spearman correlations are calculated with human vulnerability data (e.g., age and income) from the survey. Adaptation metrics (e.g., vegetation and shadow fractions) for PLRs are compared with perceived heat using Pearson correlation. Primarily, differences in perceived heat stress are explained by using the respondents USTs. Metric distributions across and within USTs are presented in violin and box plots. Spearman correlation is used to access the linkage between





USTs and human vulnerability (age and income) and Pearson correlation is used to compute the linkage between USTs, vegetation and

shadow indicators. These statistics are calculated using the USTs, TDIs with numbers assigned as indicated in Table 3.

Analysis use different administrative spatial scales, viz (Fig. 2): Boroughs, PLRs (Planungsräume/ Planning areas), and blocks. The block

scale USTs (Fig. 2b) data (e.g. grass, trees, and shadow fractions, Table 3) involves aggregating the raster data (Fig. 2).

Table 3: Data compared to survey results. The TDI uses physiological equivalent temperature (PET) values (Höppe, 1999) calculated for the
vulnerable population the number of people (#) by age group is considered. Analysis includes: fraction per Block/ PLR (grass,
trees and shadow) and percentage (%) per Block (vulnerable age groups). Summer months June, July and August (JJA). Data
availability given in Iqbal et al., 2024.

| Characteristic | Method of determination | Period | Units | Data Source |
|---|---|---|---|---|
| Thermal discomfort Index (TDI) | Calculated for indigenous residents (calm hot 2015 weather) PET (14:00), nocturnal air temperature (04:00), accounting for local tree coverage and building volume, index | Hot summer day | 1: Very favourable 2: Favourable 3: Less favourable 4: Unfavourable Weighted mean Block TDI per PLR | Senatsverwaltung für Stadtentwicklung und Wohnen, 2015 GEO-NET, 2015 |
| Urban structure types (UST) | Classification based on building structure, density, open spaces, and representative building use 52 area types grouped into 16 types but are aggregated into 7 USTs classes (Table 1) | 2021 | 1: Dense closed block 2: High-rise buildings 3: Block edge development 4: Multi-family buildings 5: Closed/ semi-open block 6: Row development 7: (semi-)detached and terraced houses | Senatsverwaltung für Stadtentwicklung und Wohnen, 2021 |
| Population Density | Registered residents place of main residence in Berlin | 2022 | Inhabitants/ hectare | Amt für Statistik Berlin-Brandenburg, 2022 |
| Block age group fraction | [age group population] / [Total Block population] | 2022 | Population (% per block) | Amt für Statistik Berlin-Brandenburg, 2022 |
| Vulnerable age groups | ≥ 65 years; ≤ 5 year (Meade et al., 2020, Dialesandro et al., 2021) | 2022 | Population (% per ring). Block centroid within a ring included | Amt für Statistik Berlin-Brandenburg, 2022 |
| Plan area fraction of grass | 1 m land cover data (2021) aggregated to 10 m to compare summer 2022 state using 10 m normalized difference vegetation index (NDVI from Sentinel-2) (Mitraka et al., 2017) | Cloud free images every 3 days JJA 2022 (54 images) | Fraction per block/ PLR Block fractions use 10 m pixels for centroids within a block boundary but not in a building footprint | Copernicus Sentinel-2 (Drusch et al., 2012) |
| Plan area fraction of trees | Same data as grass | Same data as grass | Fraction per block/ PLR | Geoportal Berlin (2022a, 2022b) Copernicus Sentinel-2 |
| Shadow fraction | Hourly shadows from buildings and trees calculated with UMEP (Lindberg et al., 2018) at 1 m pixel resolution | JJA daylight hours | Fraction per block within survey PLR, excluding building footprint | Geoportal Berlin (2022a, 2022b) and Sentinel-2 |



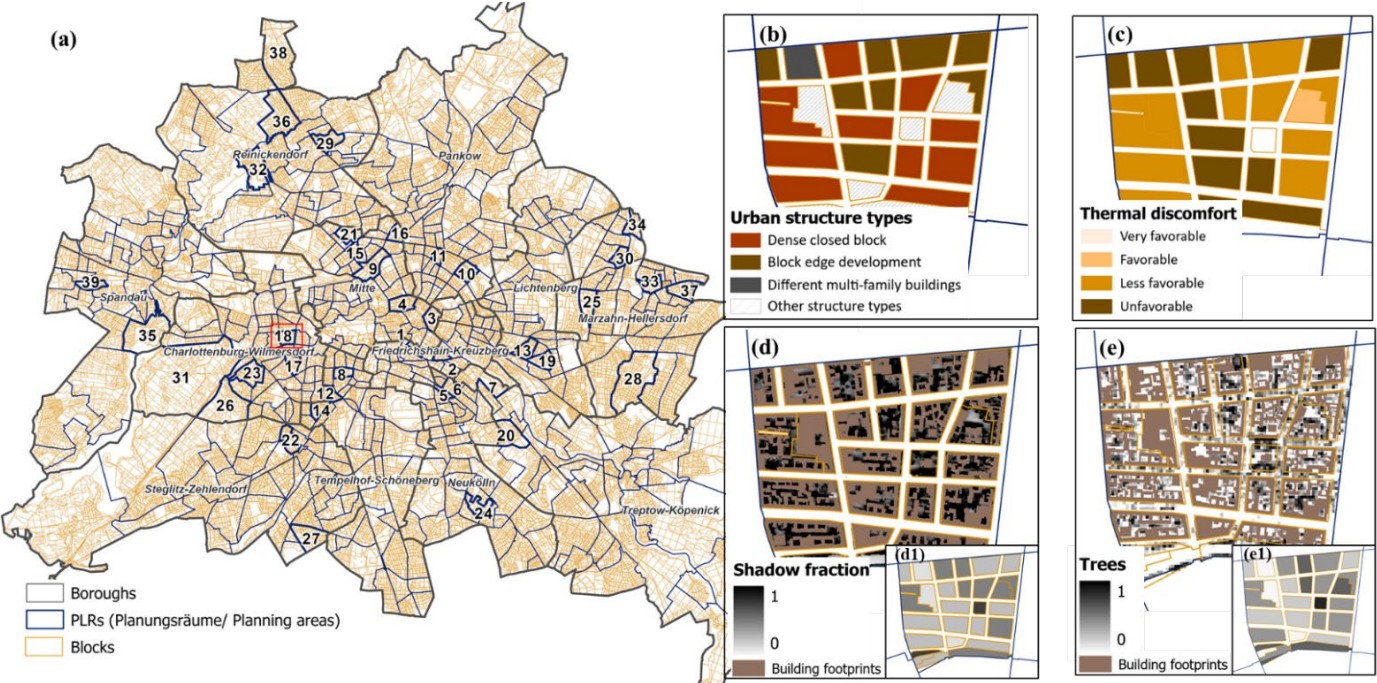

**Figure 2:** Berlin (a) administrative boundaries showing city (outer line), Boroughs (grey), PLRs (blue, planning areas) and those selected
for the household survey (numbered 1 to 39), blocks (orange) and PLR 18 (red box) for which (b) blocks and urban structure
types (UST, colour) are shown in PLR (black boundary), (c) thermal discomfort (colour), and (d) shadow fraction (0=none,
1=maximum) 1 m pixel values and (d1) block mean, and (e) plan area fraction of trees (0 none to 1 maximum) and (e1) block
mean.

## 3. Results

### 3.1. Perceived heat stress in the 38 PLRs and comparison with thermal condition

To assess perceived heat stress respondents were asked *How hot or cool do you think your neighbourhood is during a heatwave compared
to the average outdoor temperature for the city?* (Table 2, Q5.3). Across the city a greater proportion of survey respondents per PLR living
in the city centre (ring A, Fig.1) perceive more heat stress in their neighbourhood than those residing further out (Fig. 3). Overall, the
perceived heat (Table 2, Q5.3) and thermal discomfort (Table 3, TDI) is higher in the ring A PLRs than those in rings B1 and B2. In ring A
PLRs (e.g. Mitte and Friedrichshein-Kreuzberg boroughs, Fig. 3a) 76% of residents responded that their neighbourhood is hot to very hot
compared to 52% and 33% in rings B1 and B2, respectively (Fig. 3b). The differences in perceived heat between rings vary with distance
from the city centre (Fig. 3c). However, some PLRs, (e.g.) in the eastern borough of Charlottenburg-Wilmersdorf in the ring B1 (Fig. 3a,
e.g. PLR 17, Fig. 2) where 83% of respondents and Marzahn-Hellersdorf in the ring B2 (Fig. 3a, PLR 34, 37, Fig. 2) where 50 % of
respondents (Fig. 3b) perceive their neighbourhood to be hotter than the city average temperature during a heat wave event (Table 2,
Q5.3).

Respondents in other PLRs at similar distances from the centre in different parts of the city indicate different perceived heat levels (Fig. 3;
e.g. PLR 30 and 35 in ring B2; and 22 and 25 in ring B1). This also occurs in the TDI (Fig. 3). PLR perceived heat (Table 2, Q5.3) and
TDI (Table 3) are positive correlated (r ≥ 0.34, N=38). A poorer correlation is found in ring B1, which may be related to larger areal





extents of these PLRs and/or low participants number for some urban structural types (USTs). To understand this, UST (e.g. dense block and high rise), socio-demographic profiles (e.g. age, income) and adaptive capacity (e.g. access to or availability of green spaces and shadows) are explored in the following sections.

**Figure 3:** Berlin results for 38 PLRs showing (a) responses to perceived heat Q5.3 (Table 2), thermal discomfort index (TDI, Table 3), (b)
block weighted mean thermal discomfort index (TDI, Table 3) per PLR (brown, left axis) with 0 indicating all blocks 'very



favourable' and 1 all blocks unfavourable percentage of respondents indicated their perceived heat to be slightly hot to very hot (red, right axis, Table 2, Q5.3) PLR number (#) (Fig. A.1) with distance from city centre and (c, d) violin distribution with median (diamond) of ring (c) perceived heat responses and (d) TDI.

### 3.2. Perceived heat stress and USTs

Respondents report higher perceived heat stress (Table 2, Q5.3) when living in the more dense USTs than less dense USTs (Fig. 4), with median perceived heat decreasing from: dense closed blocks (hot), high-rise buildings (hot), block edge development (no difference), multi-family buildings (no difference), closed/ semi-open blocks (no difference), row development (no difference), and (semi-)detached and terraced houses (cool).

In the dense and closed blocks 67% of the respondents perceive they are living in slightly hotter to very hot conditions relative to average,

and 56% of those living in high-rise buildings indicate slightly hotter to very hot condition than average. Whereas those living in the closed/semi-open blocks (95%) and row development with green strips (87%) UST perceive they are in cool to hot conditions. In (semi-)detached and terraced houses, 63% of residents perceive their neighbourhood is slightly cooler to very cool during a heat wave compared to the average outdoor temperature for the city. It is important to note that not only USTs but also their location influence perceived heat stress; e.g., 42% of dense and closed block development is in the ring A (Fig. 1b) where 76% of residents responded that their

neighbourhood is slightly hot to very hot (Fig. 3c).

The Spearman statistic test (N=558) indicates significant correlation between USTs (ordered as Table 3) and heat perception (r=0.33 and p=<0.001).

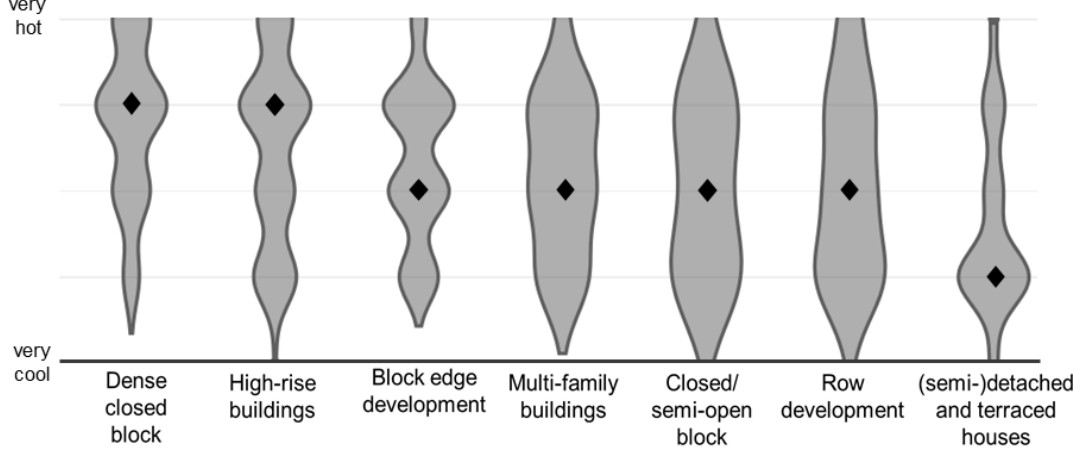

**Figure 4:** Distribution within UST of respondents perceived heat of neighbourhood (Table 2, Q5.3) ordered by decreasing median
230       (diamond).

### 3.3. Human vulnerability and adaptive capacity

#### 3.3.1. USTs, vulnerable age groups and heat perception

Many studies (e.g., Meade et al., 2020, Dialesandro et al., 2021) identify elderly (≥ 65 years) as an age group vulnerable to heat stress due to underlying health conditions influencing heat related risks. Statistically, more ≥ 65 year olds (Table 2, Q14.1) live in (semi-)detached

and terraced houses and high-rise buildings UST (Fig. 5a). However, across Berlin the block scale percentage of ≥ 65 years differ, both



within and between USTs (Fig. 5b). Overall, more live in (semi-)detached and terraced houses (median: 26%), followed by high-rise buildings (median: 25%) and row development (median: 22%). A relatively lower proportion live in multi-family buildings and block edge development (median is < 20%). Dense blocks are where elderly residents are least likely to live (median: 10 %). A Spearman correlation between the percentage of elderly (≥ 65 years) and USTs (order given in Table 3) in Berlin has a r = -0.541 (p ≤0.001).

Spatial differences are also evident between the rings by age groups (Fig. 5c). Elderly people mostly live in rings B1 and B2, between 6 and 18 km of the centre (Fig. 5c). In ring A only 13% of the total population are elderly, this increases to 22% (ring B1) and 23% (ring B2, Fig. 6b) in the outer rings. In the ring A, elderly people are most frequently living in high-rise buildings, whereas they more frequently live in detached and row houses in rings B1 and B2.

There is a weak correlation (r = 0.086, p≤ 0.004) between perceived heat and the eight age groups (Table 2, Q#14.1). This may be linked
to 43 % of the respondents aged 25 to 64 years report experiencing both high to very high heat due to commuting and spending relatively more time outside. This work-age group tends to live in the urban centre and have high exposure to heat stress. Whilst 61% of the ≥ 65 respondent group, report both a high to very high perceived heat and more heat-related health issues (Table 2, Q#5.9–5.16), with more (35.5%) very often experiencing cardiovascular health issues due to heat.





**Figure 5:** Berlin population who are 65 years or older living in different (a) USTs, (b) block scale (colour, percentage) and (c) by ring for three age groups (colour). Data source and methods: Table 3.

### 3.3.2. USTs, income and heat perception

Income plays an important role in people's adaptation capacity for challenges exacerbated by climate hazards (e.g., Abrahamson et al., 2009; Hass et al., 2021). Household monthly net income (Table 2, Q#17.8) clustered by UST (Fig. 6) shows most households living in high rise buildings, block developments and multi-family buildings have incomes close to the overall median (2900–3999€ monthly) of those surveyed. However, 25% of surveyed households in high-rise buildings and 24% in dense closed blocks said their net income is less than 2000€ monthly. Those living in (semi-)detached and terraced houses have the highest median (4000–4999€). 38.5% of respondents in



this UST have monthly net incomes ≥5000€. Whilst, in the dense closed blocks 27 % report a monthly net income ≥ 5000€ and largest interquartile range (IQR) is for 2000–5999€, indicating households from many different income groups live in this UST (Fig. 6).

Spearman correlation between USTs (order given in Table 3) and household income is weak (r = 0.22) but significant (p = <0.001). There is a weak negative (r = -0.15) correlation, but statistically significant (p ≤0.001) between household income and perceived heat; i.e. higher incomes are correlated with lower perceived heat stress. This appears conceptually logical as higher adaptive capacities are expected in wealthier households (Laranjeira et al., 2021). With 37% of surveyed households with net monthly income ≥5000€ indicating they had an air conditioning system, the results indicate a relationship between adaptive capacities and available financial resources.

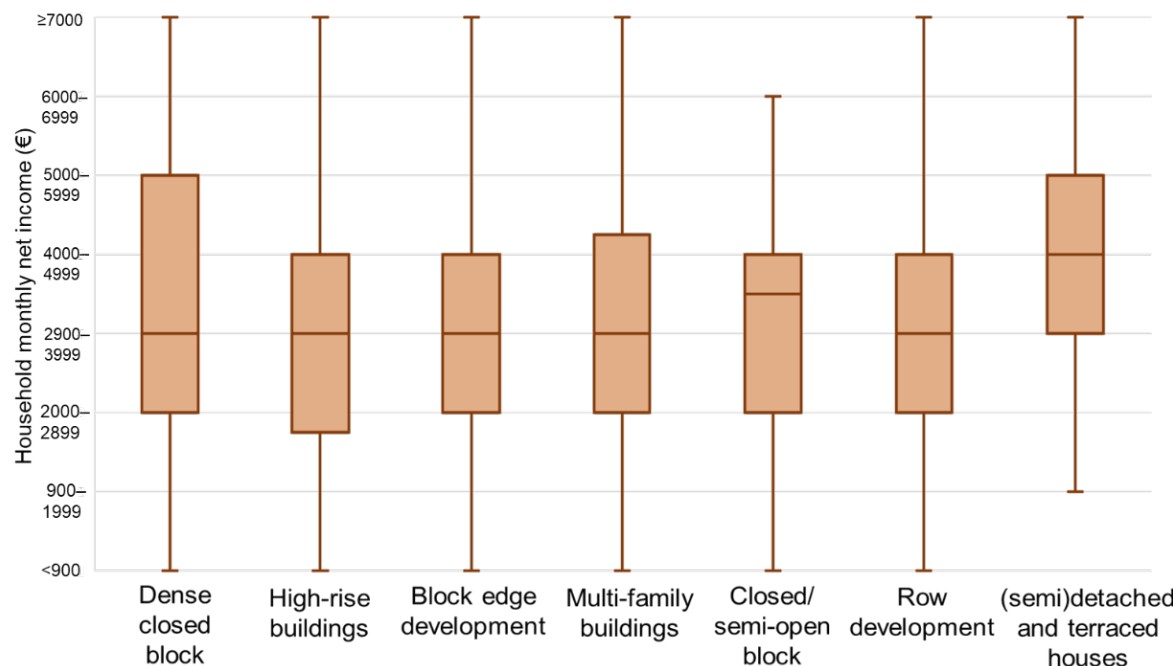


**Figure 6:**  Monthly net household income (Table 2, Q#17.8) by UST showing median (line), IQR (box) and minimum and maximum values (whiskers). Note Y axis are nonlinear classes.

### 3.3.3. USTs, availability of vegetation and heat perception

Urban vegetation can support heat stress adaptation by offsetting or buffering the adverse heat impact (Marando et al., 2022; Schwaab et
al., 2021). The plan area fraction of grass and trees is estimated using summer 2022 Sentinel-2 10 m pixel NDVI values excluding building footprint, with local both 1 m resolution land cover and tree height (Geoportal Berlin 2022a, 2022b) used to compute values for all USTs across Berlin (Table 3, Fig. 2).

The grass to tree fraction differs between USTs (Fig. 7) from similar (e.g. high-rise buildings, row), to higher fraction of trees than grass, and the reverse of higher grass fractions (cf. trees) (e.g. (semi-)detached ad terraced). The overall median fractions (diamonds, Fig. 7) also
vary with (semi-)detached and terraced houses have comparatively high fractions of both grass (0.37) and trees (0.23), followed by row development (grass: 0.27, trees: 0.28) and large estate buildings (grass: 0.23, trees: 0.25), and multi-family buildings (grass: 0.20, trees median 0.10). Dense closed blocks have very low fraction of grass (0.04) and trees (0.13) amongst the other USTs. The correlation




between fraction of vegetation and USTs (order given in Table 3) is significant (p=0.01) with a correlation coefficient of 0.778 which denotes higher association between USTs and vegetation fraction.

As the vegetation fraction is one of the six properties used to delineate the rings (Fenner et al. 2024), there is less vegetation in ring A where predominantly block structures exists. The outer rings have more vegetation and less building volume (Fenner et al. 2024, their Fig. 2) where the share of (semi-)detached and terraced houses is higher (Fig. 1b). A statistically significant correlation (p<0.001) between availability of green (Q#9.1) and perceived stress (Q#5.3) survey results is found with a correlation coefficient of 0.29.

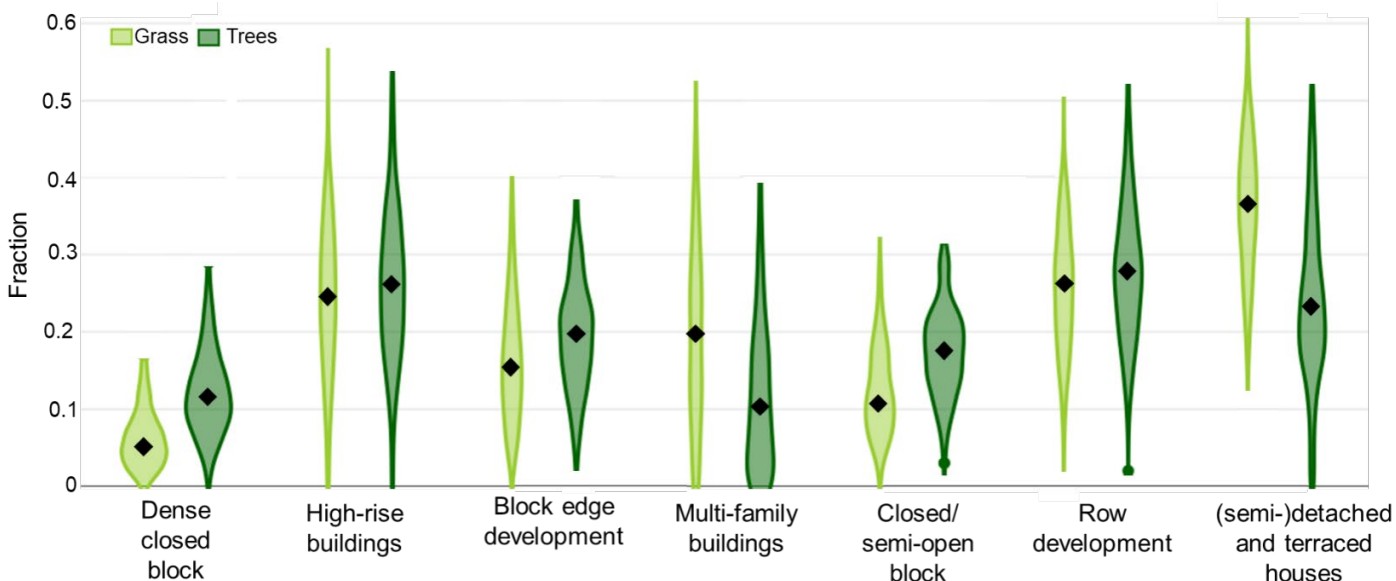

**Figure 7:** Inter-block variation and median (diamond) in grass and trees fraction (colour) by urban structure type (UST) with (Data source and method: Table 3, Fig. 2).

### 3.3.4. USTs, availability of shaded spaces and heat perception

Shading from, for example trees and buildings, are well known to create cooler areas (e.g., Lindberg and Grimmond, 2011; Bäcklin et al., 2021; Turner et al., 2023). The shadow fractions from buildings and trees are calculated for daylight hours for each summer day (June,
July, August) for each block in the surveyed PLRs (Table 3, Fig. 2).

The lowest median shadow fraction across the different USTs (Fig. 8) is for dense closed blocks (0.36), consistent with low fraction of trees (Fig. 7). The large estate high-rise buildings have one of the highest median shadow fractions (0.61) linked to the tall buildings and the presence of trees in this UST (Fig. 7). Shadow fractions are highest in row development with landscape green strips (median: 0.63) and (semi-)detached and terraced houses (median: 0.61).

Large variations of shadow fraction occur between and within USTs. The greatest variability occurs within the multi-family building UST (IQR= 0.26) followed by (semi-)detached and terraced houses (IQR=0.15). Median shadow fraction by rings for the surveyed PLRs increases from 0.43 in ring A to 0.61 ring B2, which is linked to increase in trees cover. Pearson correlation between USTs (order given in Table 3) and shadow fraction is strong and significant (r=0.55 and p=<0.001), i.e. increasing with more shaded fraction per USTs. From





the survey data, a significant (p<0.04) correlation coefficient of -0.33 is found between shadow fraction and perceived heat, indicating

reduced perceived heat stress with greater shadow fraction. Again, conceptually consistent with the expectations.

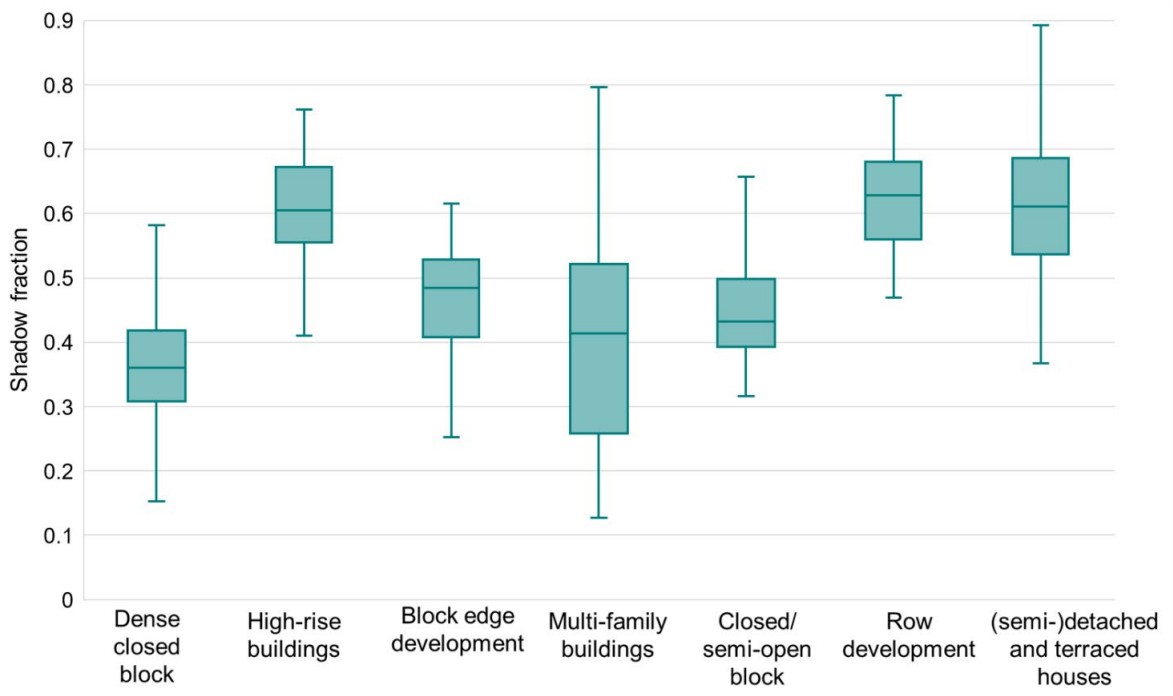

**Figure 8:** Summer (June, July, August) shadow fraction by UST with variability between blocks showing median (line), IQR (box) and minimum and maximum values (whiskers) (Data source and method: Table 3, Fig. 2).

**4.   Discussion**

Our assessment of perceived heat stress with urban structure types (USTs), people's age and income and neighbourhood location relative to the city core of Berlin demonstrate heat stress and adaptive capacities are perceived differently in various USTs and city rings.

Simplifying the city to three rings, we find a significant correlation between measured thermal discomfort and inhabitants perceived heat stress with distance from the centre of the periphery of Berlin (i.e. reducing from ring A→B1 → B2). These results are consistent with

those in Munich, Germany across city gradient (Heldens et al., 2013).  In ring A, 76% of respondents report slightly high to very high heat stress, and in ring B2 nearly a third of the respondents still report high to very high heat stress. Our analysis finds the UST people reside in is correlated with their perceived heat response. In ring B2, high perceived heat stress occurs in high-rise buildings particularly in the borough of Marzahn-Hellesdorf and multi-family buildings in Reinickendorf. Although high-rise buildings occur in all three rings, the inner ring (A) is generally more densely built-up with larger building volume (Fenner et al. 2024). 42% of dense closed block structures in

ring A having lack of the availability of vegetation and shadow accounts for climate adaptation. Thus, urban renewal projects and urban development concepts need to address both the climatic conditions within the inner city and the protection and development of green space and shaded areas within these districts where certain USTs e.g., high-rise buildings occur.





Across USTs, differences in perceived heat stress exist, as do different age groups. Notably, the elderly population have a high tendency to live in (semi-)detached and terraced houses, row development and high-rise buildings, particularly in ring B1 and B2. Given age-related

susceptibility and heat-related health problems (Sect. 3.3.1), this is a vulnerable population need to be addressed in the outer-city (e.g. ring B1 and B2). Although these households often live in single family homes, high-rise and multi-family buildings, with access to (shaded) green space, additional urban adaptation strategies could improve the demographic mix within these areas. Interestingly, with fewer elderly (16%) in ring A, it means the younger population groups are more exposed to heat stress in the inner city. Their better physical and health, should buffer them. Consequently, different urban adaptation strategies are needed for the various USTs, but also should consider

location (inner/outer city ring) as well as social composition. Differential adaptive capacities between different USTs should inform the next generation of urban adaptation plans.

Overall, the integrated analysis and assessment undertaken shows that not only the exposure to heat stress matters for urban adaptation, but also socio-demographic composition, including the consideration of differential adaptive capacities in terms of access to shaded green space and economic circumstances (e.g., income) need to be adequately considered. Particularly, areas of high concentration of elderly and

challenging socio-economic conditions (e.g., high concentration of lower income groups) require planned adaptation and support for adaptation. While elderly wealthier households in single family homes may be able to afford private adaptation measures to reduce heat stress, such as air-conditioning, the elderly population living in high-rise and multi-family buildings in the periphery (e.g., Marzahn-Hellesdorf and Reinickendorf) needs more attention. Rocha et al., 2024 in their studies also found environmental injustice in terms of lack of access to green cooling areas for vulnerable population in 14 major European urban areas. Therefore, firstly, urban development policy

should address the aging population process with urban development policies. Secondly, socio-economically disadvantaged groups and elderly living in more dense urban structures, such as high-rise buildings, typically do not have access to private green space particularly in inner urban areas. Therefore, public planning policies need to ensure that with increasing densification green space quality and access need to be secured for those group who do not to live in a house with a garden. This may be easier in large estate and high-rise buildings in the outer city region, but in both ring A and B1 such USTs exist which requires attention in the adaptation.

Finally, our characterization of urban form through USTs and city rings to capture intra-urban variability of perceived heat, human vulnerability and adaptive capacity provides an interesting insight. Beyond studying urban gradient across city rings, our approach allows a detailed study on spatial variability at neighbourhood (block) scale within the rings by an introduction of USTs. This integrated assessment approach of urban form with social fabric provides additional information on more specific adaptation requirements. It should be noted that we analysed the human vulnerability in the USTs only connected to residential uses. Working population especially those

working outside and their vulnerability is not addressed in this study due to lack of data e.g., about working conditions. Secondly, differentiation between private and public green spaces across UST and city rings is also not captured which can influence heat stress perception (Sousa-Silva and Zanocco, 2024). Nevertheless, we suggest that the linkages between USTs, vulnerable population and their differential adaptation capacities across city rings should be tested in other cities as well which can facilitate inter-city comparative studies. Role of the city size, physical and social composition, typography and climate cannot be ignored in terms of the transferability of results of

this study to other cities.



## 5.    Conclusion

In this study, we take a multi-dimensional approach combining perceived heat with urban morphology and socio-economic structure that provides essential information for enhancing adaptation towards heat-stress. This approach is based on (1) incorporating the social dimension, currently not sufficiently addressed in climate adaptation, (2) identifying the characteristics of USTs which support social

structures, and (3) employing quantitative methods to study social and physical structures across city gradient. Together they help recommendations for future climate adaptation plans to be drawn, considering the physical and social fabric of the city. This approach is exemplified in the city of Berlin. The findings show that perceived heat exposure decreases with the distance to the urban centre, however, human vulnerability and adaptive capacities depend stronger on inner variations in and differences between USTs. Therefore, USTs matter and can be linked with demographic and socio-economic information for assessing aspects of exposure, human vulnerability and adaptive

capacity.

Although UST focus on the physical structure, a deeper understanding is obtained by coupling this with socio-economic structures, human vulnerability and adaptive capacities where statistically significant correlation are found. The analysis indicates a heterogeneity in perceived heat stress and vulnerability profiles within and amongst USTs. Combined this should help identify specific local adaptation needs to be addressed in future risk management strategies in civil protection and strategic urban planning. However, urban planning

responses to climate change also require a better understanding of dynamic exposure patterns (e.g. day and night) and vulnerability. Moreover, heat-related aspects at various places e.g., in houses/apartments, in the city centre, during work and school, and while commuting need to be captured more precisely (e.g. Hertwig et al. 2024, McGrory et al. 2024). Combining people's behaviours through dedicated surveys need to be investigated and integrated into climate adaptation plans.

### Acknowledgements

This work is part of the *urbisphere* project (www.urbisphere.eu), a synergy project funded by the European Research Council (ERC-SyG) within the European Union's Horizon 2020 research and innovation programme under grant agreement no. 855005. The authors gratefully acknowledge the support of Elke Plate from the Senate Department for Urban Development, Building and Housing (Senatsverwaltung für Stadtentwicklung, Bauen und Wohnen) as well as the Population Registration Office (Melderegister der Stadt Berlin) of the city of Berlin for providing address data for conducting the household survey. Special thanks also to all participants of the household survey in Berlin

who gave their time to fill in the questionnaire.

### Author contributions

NI, MR, GS, JB, SG and DH conceptualised the study. NI, MR, and ZM curated the data. NI, MR, JB and SG developed analysis methodology. NI performed the analysis, with visualisations and drafted the manuscript.  NI, MR, JB. SG, DH, ZM and NC wrote and revised the manuscript.

### Competing interests

The contact author has declared that none of the authors has any competing interests.



## 6. Appendix: Additional Information

Table A1: Criteria used to aggregate Berlin's UST, with the 5 - 95 percentile range given.

| Characteristics | Source | Large estate with high-rise buildings | Dense block develop. closed rear courtyard | Closed block develop. rear courtyard | De-cored block-edge develop. post-war gap closure | Block-edge develop. with large quadrangles | Closed & semi-open block develop. decorative green strips & garden courtyard | Parallel row buildings with architectural green strips | Free row development with landscaped residential greenery | Row houses and duplex with yards | Densification in single-family home areas. | Detached single family houses with gardens | Villas with park-like gardens | Rental-flat buildings |
|---|---|---|---|---|---|---|---|---|---|---|---|---|---|---|
| # storeys | Geoportal Berlin, 2023 | 4.1–10.9 | 4.5–5.6 | 3.7–5.6 | 3.3–6.1 | 2.6–5.0 | 2.7–4.8 | 2.0–4.3 | 2.3–5.6 | 1.0–2.9 | 1.4–3.3 | 1.1–2.2 | 1.4–3.0 | 1.0–6.6 |
| # respondents | Household survey 2022 | 97 | 27 | 56 | 65 | 98 | 20 | 17 | 52 | 45 | 4 | 24 | 7 | 50 |
| Building age | Geoportal Berlin, 2016 | 1960s–1990s | 1870s–1918 | 1870s–1918 | after 1945 | 1920–1940 | 1870s–1918 | 1920s–1930s | 1950s–1970s | Un-specified | 1870s–present | Un-specified | 1870s–1945 | 1990s–present |
| Inhabitants/ha | Amt für Statistik Berlin-Brandenburg, 2022 | 136–479 | 263–681 | 184–594 | 152–505 | 118–423 | 99–404 | 68–320 | 68–296 | 20–132 | 33–143 | 20–68 | 14–115 | 56–434 |
| Green volume number [m³/m²] | Geoportal Berlin, 2020 | 0.6–5.9 | 0.8–3.0 | 0.9–3.9 | 1.0–5.1 | 1.2–5.4 | 1.3–4.7 | 1.6–8.8 | 1.8–7.2 | 0.2–6.1 | 1.4–8.2 | 0.7–6.7 | 1.7–8.3 | 0.1–5.5 |
| Degree of sealing [%] | Geoportal Berlin, 2021 | 31.2–63.0 | 78.3–91.4 | 64.9–89.6 | 51.3–84.6 | 40.6–72.9 | 46.4–82.2 | 29.7–62.0 | 27.0–58.6 | 21.0–50.1 | 25.3–50.1 | 21.4–40.5 | 20.1–49.5 | 33.8–84.6 |
| Floor space index | Geoportal Berlin, 2019 | 0.72–2.34 | 2.44–3.76 | 1.51–3.44 | 0.98–2.90 | 0.64–2.24 | 0.68–2.45 | 0.30–1.56 | 0.37–1.47 | 0.09–0.68 | 0.22–0.82 | 0.12–0.40 | 0.16–0.72 | 0.00–2.64 |
| Floor area ratio | Geoportal Berlin, 2019 | 0.12–0.36 | 0.53–0.72 | 0.39–0.68 | 0.29–0.65 | 0.23–0.51 | 0.25–0.58 | 0.13–0.39 | 0.14–0.35 | 0.10–0.30 | 0.15–0.30 | 0.11–0.23 | 0.11–0.29 | 0.00–0.64 |
| Satellite view | Senatsverwaltung Stadtentwicklung und Wohnen, 2020 | | | | | | | | | | | | | |
| Building block plan | Senatsverwaltung Stadtentwicklung und Wohnen, 2020 | | | | | | | | | | | | | |
| 3D view | Senatsverwaltung Stadtentwicklung und Wohnen, 2020 | | | | | | | | | | | | | |



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
