# Peer review of "How does perceived heat stress differ between urban forms and human vulnerability profiles? – case study Berlin"

_EGUsphere, 2024_

## Author Comment (AC1)

**Referee 1**

The authors present an interesting and relevant study on the interrelation of urban structure types and heat stress in the city of Berlin, more precisely, the difference between objectively measurable heat stress and the individually perceived heat stress. They base their analysis on spatial data processing, and statistics on population and to the different neighbourhoods and districts from authorities and questionnaires. The results are novel and the quality is high, but partially explained not well enough for readers without previous knowledge on the city or the topic. Most of my comments address clarification of points which raised questions during the reading. I encourage the authors to invest time in making this an excellent paper.

Dear referee,

Thank you for taking the time to read our article and for your positive and constructive review. We truly appreciate the specific issues you highlighted, along with your valuable suggestions. Attached, you will find our detailed responses and a plan for revising the manuscript in accordance with your individual comments.

The author team

| # | Comments by referee | Response | Proposed changes in the manuscript (in blue) |
|---|---------------------|----------|----------------------------------------------|
| **Abstract** | | | |
| 1 | I recommend removing the urbisphere project from the abstract as it is not relevant in this context | Thanks for your suggestion. The interdisciplinary nature of the urbisphere project is a key reason for some analysis particularly, the selection of the rings structure – so we believe it is important to retain reference to the project. Secondly, it underscores that this paper is part synergistic research between disciplines to address cities with different characterisations and their future climates, hence needing to cross international and disciplinary boundaries. Please see the revised abstract provided in response to comment 2. | |
| 2 | The abstract does not equally address problem statement, methods, results, and outlook, and largely focuses on the overall idea. Furthermore, it does not give much numeric results of which many were generated in the study. I recommend using the detailed summary from lines 352-360 in the abstract to present concise results for readers already and select some of the most relevant numbers. | We have modified the abstract as suggested and included more of the main findings (the revised version is provided in the column 3). | *Urban areas in all world regions are experiencing increasing heat stress and heat-related risks. While in-depth knowledge exists in terms of the urban heat island effect and increased heat stress in cities in the context of climate change, less is known about how individual heat perceptions and experiences differ between urban forms or with different vulnerability profiles of exposed people. It is crucial to identify and assess differences within cities relating to urban form and social structure, as both need to be considered when designing adaptation plans for heat-related risks. Here, we explore linkages between urban structure types (USTs), heat stress perception and different socioeconomic group's experiences in Berlin using a household survey, statistical and earth* |

| | | | |
|---|---|---|---|
| | | | *observation data. Our approach (1) quantifies perceived heat stress across USTs, considering characteristics such as, age, income, vegetation cover and shadow; (2) characterises social dimensions of UST to enhance it being addressed in climate adaptation; and (3) benefits from the synergistic disciplinary approach of the urbisphere project with rich social and physical datasets. Although heat stress exposure is higher in the inner-city, we find a higher percentage of vulnerable groups in the outer city (6 to 18 km from the city centre) where 78% of Berlin's elderly live. Attention is needed in climate adaptation plans based on UST to human vulnerability profiles and adaptive capacities. For example, residents of densely spaced building blocks have low median fractions of vegetation (grass: 0.04; trees: 0.13) and higher perceived heat stress (67% respondents perceive slightly hotter to very hot conditions relative to average) and very low). Our findings and methods can inform future adaptation strategies in other cities, notably to consider the different vulnerability profiles and adaptive capacities within and between USTs.* |
| 3 | The aims of the study remain unclear in the abstract: is it to quantify perception, is it the difference between vulnerability and exposure? It becomes clearer in the study itself, but I would welcome a clear statement what is your highest aim with the presented study. | We have clarified the study aim in the abstract (s*ee reviewer 1/response 2 (**R1/r2**)).* | |
| | **1. Introduction** | | |
| 4 | As you mention the increase of global air temperatures, you might add the proportionally larger increase within cities from selected references to put your study into this context | We agree and have added the it in the revised manuscript from lines 29-30. | *Cities are potentially subject to twice the levels of heat stress as compared to their rural surroundings under all representative concentration pathways (RCP) scenarios by 2050 (Wouters et al., 2017).* |
| 5 | The first part largely is based on IPCC references which are already a collection of research. It is suitable to mention the importance of the topic, but you could guide the reader to the original studies a several points to acknowledge their contribution | We now cite the original authors:

| Line | Old | New |
|---|---|---|
| 28 | IPCC, 2021 | Rosenzweig et al., 2018 |
| 31 | IPCC, 2022 | Song et al., 2016; Li et al., 2015 |
| 31 | IPCC, 2022 | Park et al., 2015 | | Li, T., Ban, J., Horton, R. M., Bader, D. A., Huang, G., Sun, Q., and Kinney, P. L.: Heat-related mortality projections for cardiovascular and respiratory disease under the changing climate in Beijing, China, Scientific reports, 5, 11441, https://doi.org/10.1038/srep11441, 2015.
Park, J., Hallegatte, S., Bangalore, M., & Sandhoefner, E.: Households and Heat Stress: Estimating the Distributional |

| | | But we retain: | | | Consequences of Climate Change. *World Bank Policy Research Working Paper No. 7479*, Available at SSRN: https://ssrn.com/abstract=2688377, 2015. |
|---|---|---|---|---|---|
| | | Line | Old | Reason | Rosenzweig, C., Ruane, A. C., Antle, J., Elliott, J., Ashfaq, M., Chatta, A., et al.: Coordinating AgMIP data and models across global and regional scales for 1.5°C and 2.0°C assessments, Phil. Trans. R. Soc. A., 376, 20160455, https://doi.org/10.1098/rsta.2016.0455, 2018. |
| | | 24 | IPCC, 2023 | A statement from the contribution of many working groups contribution which highlights the significance of topic in general. | Song, Y., Ge, Y., Wang, J., Ren, Z., Liao, Y., and Peng, J.: Spatial distribution estimation of malaria in northern China and its scenarios in 2020, 2030, 2040 and 2050, Malaria journal, 15, 345, https://doi.org/10.1186/s12936-016-1395-2, 2016. |
| 6 | I'm not clear about what you mean by "heat adaptation plans with marginalized people" and how this could look like. As you refer to this in the conclusion, it should be clearer how this could be achieved and by whom. | We understand your point. In introduction, we present the theoretical discourses on heat adaptation and socio-economic factors that might be relevant to different cases. That is why we talked about different vulnerable groups including marginalized groups. For further clarification, we rephrased it (line 49) in the introduction with examples.
In the conclusion, however, we indicated only the results specific to the case of Berlin where elderly was referred as vulnerable population and underscored them to be addressed in adaptation plans (see **R1/r37**). | | | *Heat adaptation plans exist that explicitly address marginalized and vulnerable populations who may live in lower-income neighbourhoods or be homeless. This can include planting trees and green corridors in prioritized vulnerable areas with less access to green spaces (e.g., Aburrá Valley city's Mayor's Office and the Metropolitan Area Medellín Colombia, 2021) and creating shady areas and cool places outdoors (e.g., awnings/ tents) for homeless people, distributing water bottles at counselling centres and day centres (e.g., Bochum Department of Social Affairs, Germany, 2021).* |
| 7 | It looks strange that the source is reduced to (Nature, 2021). I see that it is an editorial article but I personally think it should either include an author of the editorial team or refer to a study in this issue (595). | We have revised the citation of this editorial piece following academic guidelines (e.g., https://libanswers.umgc.edu/faq/44330; https://columbiacollege-ca.libguides.com/c.php?g=713274&p=5228077) on citing works without an author (as none is listed for this article). | | | The new reference is:
"Cities must protect people from extreme heat", Nature, 595, 331–332, https://doi.org/10.1038/d41586-021-01903-1, 2021.
In text: "Cities must protect people from extreme heat", 2021. |
| 8 | The structure of the introduction seems a bit odd, the first paragraphs follow after 1. and give background information but at this length (1 page) it could also benefit from its own sub-heading (1.1). Also, the other sub-chapters (1.1 and 1.2) have indifferent roles in the article, but both address the state of research. Would it make sense to separate the literature part | We modify the introduction structure to address the problem statement, state of the art, research gaps and study objectives. In this regard, the following changes are made:
1. Introduction
Defines problem statement: We retain lines 24-34 and 47-60 and move lines 35-47 to 1.1.
1.1. Urban form classification– combining urban morphology and heat characteristics | | | |

| | | | |
|---|---|---|---|
| | from the problem statement, definition of the research gap and the aims of the study? | Describes state of the art by addressing the following questions: Why is urban form classification important? What are commonly used local scale urban form classification systems used for urban heat island studies (i.e., local climate zones)? What are their advantages and limitations?
    1.2. Urban structure type (USTs): considering physical and socio-economic factors to assess cities
Gives the utility of USTs in research and practice, e.g., climate adaptation planning. Defines research gaps and introduces research questions. | |
| 9 | You carefully list questions but do not mention the methods on how you plan to answer them. It is clear that they follow later, but you could as well give a one-sentence outlook on what is to expect. | We agree and have modified line 118. | *Here, we capture similarities and differences of perceived heat, socio-economic structure and adaptive capacities across USTs and city rings and identified corelations between them.* |
| 10 | Again, highlighting the name of your research project in the last paragraph is rather uncommon and does not add relevant information. I'd say this belongs to the acknowledgments. | Some issues can only be understood when looking at the goals of the project urbisphere – e.g., the ring structure. Therefore, readers will benefit from further information when a reference to the project urbisphere is provided (see **R1/r2**). | |
| | **2. Methods** | | |
| 11 | The quotes of authorities in the text are partially very long (Deutscher Wetterdienst and Senatsverwaltung für Stadtentwicklung, 2010). Can you identify authords of the works and quote them accordingly? As an alternative, you could use abbreviations to make the citations in the text more compact. | Thank you for the suggestion. We have changed the quote to:

| Line | Old | New |
|---|---|---|
| 125-129 | Deutscher Wetterdienst and Senatsverwaltung für Stadtentwicklung, 2010 | Deutschländer et al., 2010 | | Deutschländer, T., Früh, B., Koßmann, M., & Roos, M., Wienert, U. Berlin im Klimawandel - eine Untersuchung zum Bioklima, Edited by Behrens, U.; Grätz, A. Deutscher Wetterdienst and Senatsverwaltung für Stadtentwicklung, https://digital.zlb.de/viewer//fulltext/15490747/1/. last accessed: September 02, 2023, 2010. |
| 12 | Again, you mention the urbisphere campaign, please stick to the references to existing papers. | Fenner et al. (2024) is published and presents the ring structure generally and the details for the urbisphere-Berlin campaign. | |
| 13 | It is not logic to me that you present the city as a polycentric phenomenon but use (concentric) rings to describe it. This sounds contradictive to someone who is not involved in urban studies. Please clarify. | The logic of using concentric rings and urban structure types as well as planning areas (PLRs) is based on our core interest to combine different perspectives and typologies to classify cities representing urban physical, socio-economic and climatic conditions. The concentric ring structure (Fenner et al., 2024) is one representation that allows us to combine socio-economic data e.g., elderly with questions of climatic | We added the following description in line 133 to clarify this:
*Our first premise is that there are broadly two city zones (inner and outer city), surrounded by a rural area. The proposed ring structure for Berlin is defined by an interdisciplinary team (meteorology, remote-sensing and urban/spatial planning) as an attempt to provide a simplified and comparative approach replicable in other cities (Fenner* |

| | | conditions e.g., perceived heat. It is not a contradiction to the statement that Berlin is characterized by a polycentric structure in terms of urban planning and urban development approaches. We adapted the proposed ring structure in this paper to compare the results from the climate analysis with socio-demographic structure and provide complementary approaches. See **R2/r1** for details. | *et al. 2024). Another important aim is to compare results and provide complementary methods and approaches between urban climate studies and urban planning studies. In this context also classifications and analysis schemes of different research communities are applied and linked.* |
|---|---|---|---|
| 14 | Can you briefly mention how the reduction of 13 USTs to 7 classes was performed (e.g., "based on xy") or refer to a source where it is described? The table in Appendix A1 is not clear to me. | Following description (line 136) to provide basis for reducing the number of USTs. | *We use the socio-demographic and physical data (e.g., population density, building morphology, number of storeys, building age, green volume, degree of sealing, Table A1) to characterise the USTs. Using the $5^{th}$ to $95^{th}$ percentile ranges of the 13 classes we can reduce this to seven classes (Fig. 1a, Table 1).* |
| 15 | The source to Senatsverwaltung (2021) does not explain how the USTs were delineated (data source, criteria, methods…) for Berlin, is there any information available? | We add reference to Senatsverwaltung für Stadtentwicklung und Wohnen (2020) as it provides detailed documentation of the criteria, data basis and method used for the delineation of USTs. | Senatsverwaltung für Stadtentwicklung und Wohnen: Dokumentation Bodennutzung und Stadtstruktur 2020, https://www.berlin.de/umweltatlas/_assets/literatur/nutzungen_stadtstruktur_2020.pdf?ts=1726132803, last accessed: 01/09/2024, 2020. |
| 16 | What do the labeled names in the map of Figure 1a represent? Maybe you can add this in the caption. Are these the Buroughs? Someone who is not from Berlin might get confused about what is a PLR and what is a Burough. | The labels in Figure 1a give the names of the city boroughs. | Updated caption: *Berlin study area (a) inner (A) and outer (B1, B2) city rings and Senatsverwaltung für Stadtentwicklung und Wohnen (2021) urban structure types (UST, Table 1) with example photos and boroughs labelled, and (b) plan area of USTs (%) in each city ring. (Photo source: Marvin Ravan).* |
| 17 | You mention 39 residential addresses in line 152 but 38 in chapter 3.1 which is confusing | Given the low number of participants responses from one PLR (No 39, 3 responses) it was removed from further analyses (line 158). To avoid confusion, PLR 39 is removed from Figure 2. | |
| 18 | Can you briefly explain the stratification process mentioned in line 153 that led to the sampling visible in Figure 2? If this is the case, please add a reference to Figure 2. | PLRs were selected by an expert group (5 persons) based on multiple criteria, e.g., heat exposure (Senatsverwaltung für Stadtentwicklung und Umwelt, 2014), population density and representation of different age groups (Amt für Statistik Berlin-Brandenburg, 2022), unemployment levels (Senatsverwaltung für Stadtentwicklung und Wohnen Berlin, 2019), heat mortality rate (Schuster et al. 2014). We selected 39 PLRs that reflect diverse socio-economic, demographic and typological characteristics while ensuring to cover all boroughs in the city of Berlin. | |

| | | | |
|---|---|---|---|
| 19 | If the survey was accessible by QR codes I suspect a bias towards the participation of elderly which are a focus group of your analysis. (How) can you be sure that the less technically affine population is included? I refer to this at a later point again with a constructive suggestion. | We posted an invitation letter to 10,000 residential addresses located in the 39 Berlin PLRs selected. The letter stated that if the respondent had technological constraints, they could ask for a printed copy of the questionnaire by phone. We posted questionnaires in response to the calls received.

It should be noted that around 27.2% (N=155) of respondents are classed as "elderly" (65 and older) (see **R1/ r34** for details). | |
| 20 | I would welcome a flowchart which explains how different data sources, methods and information flow together in your analysis | A flow chart explaining the data sources, methods and analyses will be added to the Appendix. | |
| 21 | Table 2: Colum Q# indicates that the actual questionnaire was significantly larger. Please shortly explain the total size and topics and why you chose the 7 in your study. | Description added/modified in section 2.2, line 159. | *The survey has questions on household's heat stress perception and experience, living conditions (e.g., USTs, building information, green space access), mobility, early warning system, coping measures adaptation options, and socio-demographic characteristics (e.g., age, income, education and working condition) (Table 2). Given the focus of this paper, we utilise seven survey topics related to perceived heat and physical and socio-economic factors of people living in different USTs.* |
| 22 | Lines 170-176: As you are describing how the results of your analysis will look like, please refer to each of the following sub-chapters where the correlations and plots are to be expected. | We will add cross-references to the sub-sections where plots and correlations are presented. | |
| 23 | Table 3: The table mentions 16 UST but in your paper 13 are mentioned as well, this is confusing | Table 3 corrected, thanks for pointing this out. | |
| 24 | Table 3: Parts of the captions "Analysis includes […] in Iqbal et al, 2024" this could be put I the text to make the caption more compact | Parts of the caption "Analysis includes: fraction per Block/ PLR (grass, trees and shadow) and percentage (%) per Block (vulnerable age groups)." is shifted to the text. However, we keep "Data availability given in Iqbal et al. 2024" in the caption to refer to the source where the processed data used for this analysis is available. Summer months June, July and August (JJA) is also retained in the caption. | |
| 25 | Table 3: I am not clear about the term indigenous residents (TDI) | The term indigenous residents is removed. | |

| 26 | Table 3: "Copernicus Sentinel-2" seems redundant as a source if the quoted reference contains the actual data source and method, I'd prefer an either citation-based source or the consistent use of the underlying geospatial dataset (probably more complicated to add for all parameters). | Data sources column revised to data reference.

| Characteristic | Old | New |
|---|---|---|
| Plan area fraction of grass | Copernicus Sentinel-2 | Drusch et al., 2012 |
| Plan area fraction of trees | Copernicus Sentinel-2 | Drusch et al., 2012 |
| Shadow fraction | Sentinel-2 | Gasco et al., 2014 | | Drusch, M., del Bello, U., Carlier, S., Colin, O., Fernandez, V., Gascon, F., et al.: Sentinel-2: ESA's Optical High-Resolution Mission for GMES Operational Services. Remote Sensing of Environment, 120, 25–36. https://doi.org/10.1016/j.rse.2011.11.026, 2012.
Gascon, F., Cadau, E., Colin, O., Hoersch, B., Isola, C., López Fernández, B., et al.: Copernicus Sentinel-2 mission: products, algorithms and Cal/Val. 9218, 92181E. https://doi.org/10.1117/12.2062260, 2014. |
| 27 | Figure 2: The boroughs, PLRs and blocks are not clear, why not showing the USTs for the entire city as this is your main unit of analysis? | USTs for the entire city are presented in Figure 1a. Figure 2a is modified to make block, PLR and borough boundaries clearer. | |
| | **3. Results** | | |
| 28 | As mentioned earlier, the 38/39 mismatch of PLRs must be resolved | See response **R1/r17**. | |
| 29 | How were PLRs which range between several kilometers assigned to one interval? | The PLRs are assigned to the ring in which they have their centroids to correlate perceived heat with thermal discomfort index (TDI) as presented in section 3.1. We explored prominent differentiation of USTs (Fig. 1b) and perceived heat stress (Fig. 3 (c and d)) in terms of rings and further provided some key factors contributing to such different patterns. Please see **R2/ r1** for details about ring structure. | |
| 30 | Figure 3: The order of a-d seems odd to me because b is at the bottom | Figure 3a and b are presenting partially the same PRL-based data spatially and statistically, respectively. Therefore, the sequence of label follows from top to bottom. | |
| 31 | Figure 3c/d, Figure 4, Figure 5a: I don't think violin plots are a legitimate choice here because the y-axis is of ordinal scale and the violin plots suggest a continuous variable. Stacked columns as in Figure 5c. | Figure 3c/d and Figure 4 are modified as per recommendation i.e., stacked columns. However, Figure 5a is based on continuous statistical data i.e., percentage of elderly population at block, therefore, violin plots are an appropriate choice in this case. | |
| 32 | Figure 3b and 3d: I suggest to name it "thermal discomfort index (TDI)" to make it clearer that this is the measure retrieved from other data then the perceived heat from the questionnaire. | Now referred to as Thermal discomfort index (TDI) – corrected throughout. | |

| 33 | Line 234: "Statistically, more >65 year olds live in semi-detached and terraced houses" → please rephrase because it indicates an absolute dominance while it is just a proportional statement. For example, "a higher share of >65 year olds lives in semi-detached and terraced houses" | Thanks for the correction. Statement will be modified as suggested. | |
|---|---|---|---|
| 34 | Chapter 3.3.1: As you are correctly addressing statistical significance, it would be good to briefly mention the sample size of these calculations again in the beginning. If these calculations are only based on the 565 responses, I would suspect that the age bias of the survey methods (as indicated above) distorts the actual conditions. Would it help to display a histogram of the age groups of the questionnaire and a histogram of total Berlin's population to proof that each group is equally represented? You could place this in the appendix to give your results more validity. | The correlation between UST and percentage of elderly (line 239) is based on statistical and geospatial datasets at block scale from the city of Berlin (Table 3). However, the correlation between perceived heat and age groups is based on survey data (N=564, Table 2). We have added the number of responses in this case.

New Figure B1 referred in line 244 shows the survey respondents and number of people Berlin by age group. |
Appendix B1: Age-group histograms of (a) survey respondents and (b) the population of Berlin from Statistisches Bundesamt, 2022 |
| 35 | Figure 4 b: Elderly people is missing a unit: "Share of elderly people (>65 years) [%]" | *Share of elderly people (>65 years) [%]* has been added to the unit of Figure 4b. | |
| 36 | Chapter 3.3.3.: I suspect that vegetation played already a role in in the delineation of the 13 USTs. If this is the case, Figure | Vegetation [e.g., green volume m³/m²] is used by Senatsverwaltung für Stadtentwicklung und Wohnen (2020) in their delineation of USTs. However, by looking at the quality | |

| | | of green, we have distinguished between grass and trees in different USTs in the analysis (section 3.3.3.) considering their different thermal impacts. Secondly, plan area fraction of grass and trees is estimated using summer 2022 data to make it consistent with the household survey implementation timeline, e.g., to correlate current supply of vegetation with perceived heat stress. | |
|---|---|---|---|
| | 7 just replicates the process of their generation and have no additional analytical value. | | |
| **4. Discussion** | | | |
| 37 | Lines 315-325 and 320-321: How should processes of urban planning and development address this call for action?

Why does it matter if elderly live in B1 or B2? I agree that vulnerable people must be protected, but your plea for action is not very concrete here. Also in line 334-335: Which options do city planners have to include the aging process in cities? Doesn't this mean that, in the end, all parts of the city are subject to demographic shifts and urbanization processes and have to be equally prepared for heat? In other words, how can knowledge about spatial patterns of thermal discomfort and heat stress actually help planning? As this is already the part of the outlook where you have to sell your study at the highest price (it does contain many new insights), who will be later using this information? | Many thanks for the insight. We agree that all parts of the cities have to be resilient to heat stress, but highly exposed and vulnerable locations and people should be prioritised in adaptation planning (line 315-325). To further clarify how can the knowledge about spatial patterns of thermal discomfort and perceived heat stress actually help planning, we have modified the line 340 in the discussion.

Secondly, line 227-335 and 343-349 highlight some of key aspects related to adaptation for the elderly. We added potential measures in line 322 to provide some examples.

Moreover, we underscore the need of dedicated studies to understand urban transition in the conclusion. | Line 340 following description is added:
*Understanding spatial patterns of thermal discomfort and heat stress is critical for targeted interventions to improve the liveability of urban areas in the context of climate change. The characterization of urban form using Urban Spatial Typologies (USTs) and city rings allows for a detailed understanding of the variability in perceived heat, human vulnerability, and adaptive capacity across different spatial scales. This nuanced approach supports more targeted interventions for urban development and climate change adaptation.*

Following modification are made in line 318-322:
Notably, the elderly population have a high tendency to live in *(semi-)detached and terraced houses [26%], high-rise buildings [22%] row development and high-rise buildings row development [18%],* particularly in ring B1 and B2. Given age-related susceptibility and heat-related health problems (Sect. 3.3.1), this is a vulnerable population need to be addressed in the outer-city (e.g., ring B1 and B2). Although these households often live in single family homes, high-rise and multi-family buildings, with access to (shaded) green space, *additional urban adaptation measures such as inclusive public and open spaces and community centres* could improve the demographic mix within these areas.

Following details are added in conclusion line 348:
*There is a need for dedicated studies to investigate demographic shifts and urbanization processes for identifying urban transformation pathways. In this respect, physical (e.g., tree growth) and social transitions (e.g., aging population,* |

| | | | *work force changes) over time need to be account for in adaptation plans.* |
|---|---|---|---|
| 38 | Line 349: Consider removing "Role of the" | "Role of the" has been removed. | |

**5. Conclusion**

| 39 | As stated above, the first paragraph would make a good abstract already. | Modified abstract (see **R1/r2**). | |
|---|---|---|---|
| 40 | Line 363: "Combined this" does not sound like a grammatically correct phrase. | "Combined this" is removed. Sentence (line 363-364) rephrased. | *Collectively, this approach will facilitate the identification of specific local adaptation needs to be addressed in future risk management strategies for civil protection and strategic urban planning.* |
| 41 | Line 367-368: I wonder how such a prioritization could look like. I think in the end all parts of the city have to be heat resilient and any attempt for local adaptation at the architectural level will be obsolete within years, as the city and its work force, elderly, etc. constantly change. But maybe I am just missing the point. | In the discussion section, we argue that not only heat exposed locations e.g., city centre but also vulnerable population e.g., elderly in the outer city needs an attention in adaptation plans (line 315-325). There are already some evidences of targeted interventions such as Berlin's "Sustainable Renewal (Das Programm Nachhaltige Erneuerung in Berlin, 2023)" programme which targets predominantly large housing estates and socially disadvantaged households and "Lively centres and neighbourhood initiatives (Lebendige Zentren und Quartiere, 2020)" aim to implement appropriate protection and adaptation measures in highly exposed areas of city centres. Particularly, we emphasised those locations that might not be highly exposed to heat but due to the vulnerability factors (e.g., related to age, income) need to also be prioritised in adaptation planning. We highlighted the need of dedicated studies to understand physical (e.g., tree growth) and social transitions (e.g., aging population, work force changes) over time in the conclusion, see **R1/r37**. | |

**6. Acknowledgments**

| 42 | There is a fullstop in the list of authors where there should be a comma ("JB. SG"). | Replaced with a comma. Thanks for pointing it out. | |
|---|---|---|---|

**7. References**

Amt für Soziales: Hitzekonzept: Obdach- und Wohnungslose bei „Hitzewellen" schützen. https://www.bochum.de/Pressemeldungen/14-Juni-2021/Stadt-stellt-Hitzekonzept-fuer-Obdachlose-vor, Retrieved December 05, 2023, 2021.

Amt für Statistik Berlin-Brandenburg (AfS) / Senatsverwaltung für Stadtentwicklung und Wohnen Berlin: Lifeworld-oriented Spaces (LOR) in Berlin, https://www.berlin.de/sen/sbw/stadtdaten/stadtwissen/sozialraumorientierte-planungsgrundlagen/lebensweltlich-orientierte-raeume/, last accessed: 2/03/2023, 2021.

Anger, B., Heinze, A., Caldow, C. City presents heat concept for homeless people: https://www.bochum.de/Pressemeldungen/14-Juni-2021/Stadt-stellt-Hitzekonzept-fuer-Obdachlose-vor (in German), last accessed: September 29, 2024, 2021.

Aburrá Valley city's Mayor's Office. Medellín Climate Action Plan 2020-2050, Municipality of Medellín: https://www.medellin.gov.co/es/wp-content/uploads/2024/03/PAC_Medellin_Libro_Digital.pdf (in Spanish), last accessed: September 29, 2024, 2021.

Bertram, R.: How "green corridors" are driving sustainable policies in Medellín: https://energytransition.org/2023/12/how-green-corridors-are-driving-sustainable-policies-in-medellin/, Retrieved September 20, 2024, 2023.

Dialesandro, J., Brazil, N., Wheeler, S., and Abunnasr, Y.: Dimensions of Thermal Inequity: Neighborhood Social Demographics and Urban Heat in the Southwestern U.S, International journal of environmental research and public health, 18, https://doi.org/10.3390/ijerph18030941, 2021.

Deutschländer, T., Früh, B., Koßmann, M., & Roos, M., Wienert, U. Berlin im Klimawandel - eine Untersuchung zum Bioklima, Edited by Behrens, U.; Grätz, A. Deutscher Wetterdienst and Senatsverwaltung für Stadtentwicklung, https://digital.zlb.de/viewer//fulltext/15490747/1/. last accessed: September 02, 2023, 2010.

Drusch, M., del Bello, U., Carlier, S., Colin, O., Fernandez, V., Gascon, F., et al.: Sentinel-2: ESA's Optical High-Resolution Mission for GMES Operational Services. Remote Sensing of Environment, 120, 25–36. https://doi.org/10.1016/j.rse.2011.11.026, 2012.

Fenner, D., Christen, A., Grimmond, S., Meier, F., Morrison, W., Zeeman, M., Barlow, J., Birkmann, J., Blunn, L., Chrysoulakis, N., Clements, M., Glazer, R., Hertwig, D., Kotthaus, S., König, K., Looschelders, D., Mitraka, Z., Poursanidis, D., Tsirantonakis, D., Bechtel, B., Benjamin, K., Beyrich, F., Briegel, F., Feigel, G., Gertsen, C., Iqbal, N., Kittner, J., Lean, H., Liu, Y., Luo, Z., McGrory, M., Metzger, S., Paskin, M., Ravan, M., Ruhtz, T., Saunders, B., Scherer, D., Smith, S. T., Stretton, M., Trachte, K., and van Hove, M.: urbisphere-Berlin Campaign: Investigating Multiscale Urban Impacts on the Atmospheric Boundary Layer, Bulletin of the American Meteorological Society, 105, E1929-E1961, https://doi.org/10.1175/BAMS-D-23-0030.1, 2024.

Gascon, F., Cadau, E., Colin, O., Hoersch, B., Isola, C., López Fernández, B., et al.: Copernicus Sentinel-2 mission: products, algorithms and Cal/Val. 9218, 92181E. https://doi.org/10.1117/12.2062260, 2014.

Li, T., Ban, J., Horton, R. M., Bader, D. A., Huang, G., Sun, Q., and Kinney, P. L.: Heat-related mortality projections for cardiovascular and respiratory disease under the changing climate in Beijing, China, Scientific reports, 5, 11441, https://doi.org/10.1038/srep11441, 2015.

Meade, R. D., Akerman, A. P., Notley, S. R., McGinn, R., Poirier, P., Gosselin, P., and Kenny, G. P.: Physiological factors characterizing heat-vulnerable older adults: A narrative review, Environment international, 144, 105909, https://doi.org/10.1016/j.envint.2020.105909, 2020.

Park, J., Hallegatte, S., Bangalore, M., & Sandhoefner, E.: Households and Heat Stress: Estimating the Distributional Consequences of Climate Change. World Bank Policy Research Working Paper No. 7479, Available at SSRN: https://ssrn.com/abstract=2688377, 2015.

Rosenzweig, C., Ruane, A. C., Antle, J., Elliott, J., Ashfaq, M., Chatta, A., et al.: Coordinating AgMIP data and models across global and regional scales for 1.5°C and 2.0°C assessments, Phil. Trans. R. Soc. A., 376, 20160455, https://doi.org/10.1098/rsta.2016.0455, 2018.

Senatsverwaltung für Stadtentwicklung und Wohnen: Das Programm Nachhaltige Erneuerung in Berlin 2023, https://www.berlin.de/sen/stadtentwicklung/_assets/quartiersentwicklung/foerderprogramme/nachhaltige-erneuerung/programm/2023-bf-ne_programmblatt.pdf?ts=1690559088, last accessed: 041/09/2024, 2023.

Senatsverwaltung für Stadtentwicklung und Wohnen: Lebendige Zentren und Quartiere 2020, https://www.berlin.de/sen/stadtentwicklung/quartiersentwicklung/staedtebaufoerderung/lebendige-zentren-und-quartiere/, last accessed: 041/09/2024, 2020.

Senatsverwaltung für Stadtentwicklung und Wohnen: Dokumentation Bodennutzung und Stadtstruktur 2020, https://www.berlin.de/umweltatlas/_assets/literatur/nutzungen_stadtstruktur_2020.pdf?ts=1726132803, last accessed: 01/09/2024, 2020.

Song, Y., Ge, Y., Wang, J., Ren, Z., Liao, Y., and Peng, J.: Spatial distribution estimation of malaria in northern China and its scenarios in 2020, 2030, 2040 and 2050, Malaria journal, 15, 345, https://doi.org/10.1186/s12936-016-1395-2, 2016.

Statistisches Bundesamt: Altersstruktur der Bevölkerung in Berlin, 2022 und 2070, https://www.demografie-portal.de/DE/Fakten/Daten/bevoelkerung-altersstruktur-berlin.csv?__blob=publicationFile&v=4, last accessed: 3/09/2023, 2022.

Wouters, H.: Heat stress increase under climate change twice as large in cities as in rural areas: A study for a densely populated midlatitude maritime region. Geophys. Res. Lett., 44(17), 8997–9007, 2017.

---

## Author Comment (AC2)

**Referee 2**

This paper examines how the perception of heat stress in Berlin differs depending on urban forms and human vulnerability profiles. The motivation behind this work - to understand drivers and perceptions of heat stress in order to inform adaptation plans - is important, and the work being done by these authors has the potential to contribute to improving heat adaptation planning.

Dear referee,

Thank you for taking the time to read our article and for your positive and constructive review. We truly appreciate the specific issues you highlighted, along with your valuable suggestions. Attached, you will find our detailed responses and a plan for revising the manuscript in accordance with your individual comments.

The author team

| # | Comments by referee | Response | Proposed changes in the manuscript (in blue) |
|---|---|---|---|
| 1 | Unfortunately, the use of rings to define perfectly circular zones radiating from the city center for analysis is arbitrary and likely masks geographic correlations that may be stronger than those found. The paper states that there are two city centers, yet the zones radiate from a single city center. Additionally, while cities tend to grow out from centers, they generally do not do so symmetrically due to geographic and historical influences. Defining zones within the city based on time of development, building types, population density, building density, development function (e.g. industrial, residential), and/or demographics would provide more logical geographic divisions to study. | The comment highlights an important challenge – how to characterize cities and linkages between the urban physical, socio-economic and climatic conditions. Since, the *urbisphere* project (see website https://urbisphere.eu/index.html) aims to explore linkages between cities and climatic conditions, we apply different typologies to characterize cities. More precisely, we use the typologies primarily used in urban planning, such as Urban Structure Types (USTs) and Planning Areas (PLRs) in Berlin and juxtaposed and linked the statistical analysis done for PLRs and USTs to the methods used for assessing and modelling urban climate conditions in a broader sense – like the developed and published ring structure (Fenner et al., 2024). The ring structure analysis zones for Berlin are defined as part of the *urbisphere* project campaign by a team from different disciplines (meteorology, remote-sensing and urban/spatial planning) as an attempt to provide a simplified comparative approach replicable in other cities (Fenner et al. 2024). The *urbisphere*-Berlin campaign analysis of form (building area fraction, vegetation area fraction, and building volume) and function data (population density, anthropogenic heat influxes) identified an inner-city ring (radius 6 km) and an outer city ring (radius 18 km) (Fenner et al., 2024, their Fig. 2). Consequently, we agree that the cities' growth is rarely symmetrical due to various geographic, historical, and social | We added the following description in line 133 to clarify this: *Our first premise is that there are broadly two city zones (inner and outer city), surrounded by a rural area. The proposed ring structure for Berlin is defined by an interdisciplinary team (meteorology, remote-sensing and urban/spatial planning) as an attempt to provide a simplified and comparative approach replicable in other cities (Fenner et al. 2024). Another important aim is to compare results and provide complementary methods and approaches between urban climate studies and urban planning studies. In this context also classifications and analysis schemes of different research communities are applied and linked.* |

| | | | |
|---|---|---|---|
| | | factors. Therefore, two prominent local spatial units i.e., urban structure types (USTs) delineated by the Senatsverwaltung für Stadtentwicklung und Wohnen (2021) and PLRs (planning areas) defined by Amt für Statistik Berlin-Brandenburg and Senatsverwaltung für Stadtentwicklung und Wohnen Berlin (2021) are used in the analysis (i.e., section 3.1., Figure 3(a) and 3(b) and Figure 4–8). Moreover, correlation analysis of the perceived heat with the survey participants are based on PLRs (section 3.1.) and USTs (section 3.2. and section 3.3), reflecting the use of diverse characterization of the city. | |
| 2 | A second methodological problem is the manner in which respondents were recruited for the study. Age of respondents is an important factor, but the use of a QR-code and an online survey would likely lead to a lower response rate from older people who are generally less accustomed to technology. | We posted an invitation letter to 10,000 residential addresses located in the 39 Berlin PLRs selected. The letter stated that if the respondent had technological constraints, they could ask for a printed copy of the questionnaire by phone. We posted questionnaires in response to the calls received. The survey population sample and household data gathered captures and represents the elderly population quite well. Around 27.2% (N=155) of respondents are classed as "elderly" (65 and older). In 2022, the elderly (age 65 and above) had a share of 19% of the total population in Berlin and this group is expected to increase over the next decades (see: Demographie-Portal online: https://www.demografie-portal.de/DE/Fakten/bevoelkerung-altersstruktur-berlin.html). We further clarified it using a histogram of surveyed respondents and Berlin's population by age group. Please see **R1/ r34** for details. | |
| 3 | The final issue with the paper is the writing. Grammar and sentence structure are problematic in many places. The organization and development of ideas is also weak in some places. For example, the second and third sentences of the introduction talk about the problem of heat in urban areas, but it is not until the second paragraph that the paper establishes that heat in urban areas is a separate problem from heat in general. A thorough edit for clarity is needed. | Thanks for the comment. The paper will be edited thoroughly to improve the development of ideas and sentence structure (e.g., please see **R1/ r6, R1/ r8, R1/ r14** etc.). Language and grammar check is also carried out by native speakers. | We added the following statement in the revised manuscript to highlight proportionally larger increase within cities from lines to establish 29-30: *Cities are potentially subject to twice the levels of heat stress as compared to their rural surroundings under all RCP (Representative Concentration Pathways) scenarios by 2050 (Wouters et al., 2017).* |

| 4 | All that said, I think the data and the analyses are on the right track. With a better geographic analysis, this paper would have great potential for publication. | Once again thank you for your constructive review. We make sure to fully address your comments in the revised paper. | |
|---|---|---|---|

**References**

Amt für Statistik Berlin-Brandenburg (AfS) / Senatsverwaltung für Stadtentwicklung und Wohnen Berlin: Lifeworld-oriented Spaces (LOR) in Berlin, https://www.berlin.de/sen/sbw/stadtdaten/stadtwissen/sozialraumorientierte-planungsgrundlagen/lebensweltlich-orientierte-raeume/, last accessed: 2/03/2023, 2021.

Fenner, D., Christen, A., Grimmond, S., Meier, F., Morrison, W., Zeeman, M., Barlow, J., Birkmann, J., Blunn, L., Chrysoulakis, N., Clements, M., Glazer, R., Hertwig, D., Kotthaus, S., König, K., Looschelders, D., Mitraka, Z., Poursanidis, D., Tsirantonakis, D., Bechtel, B., Benjamin, K., Beyrich, F., Briegel, F., Feigel, G., Gertsen, C., Iqbal, N., Kittner, J., Lean, H., Liu, Y., Luo, Z., McGrory, M., Metzger, S., Paskin, M., Ravan, M., Ruhtz, T., Saunders, B., Scherer, D., Smith, S. T., Stretton, M., Trachte, K., and van Hove, M.: urbisphere-Berlin Campaign: Investigating Multiscale Urban Impacts on the Atmospheric Boundary Layer, Bulletin of the American Meteorological Society, 105, E1929-E1961, https://doi.org/10.1175/BAMS-D-23-0030.1, 2024.

Statistisches Bundesamt: Altersstruktur der Bevölkerung in Berlin, 2022 und 2070, https://www.demografie-portal.de/DE/Fakten/Daten/bevoelkerung-altersstruktur-berlin.csv?__blob=publicationFile&v=4, last accessed: 3/09/2023, 2022.

Senatsverwaltung für Stadtentwicklung und Wohnen: Urbane Struktur / Urbane Struktur - Flächentypen differenziert, https://www.berlin.de/umweltatlas/en/land-use/urban-structure/, last accessed: 2/03/2023, 2021.

Wouters, H.: Heat stress increase under climate change twice as large in cities as in rural areas: A study for a densely populated midlatitude maritime region. Geophys. Res. Lett., 44(17), 8997–9007, 2017.